# Clinical presentation, microbiology, and prognostic factors of prosthetic valve endocarditis. Lessons learned from a large prospective registry

Antonio Ramos-Martínez[1,2,3]*, Fernando Domínguez[4], Patricia Muñoz[4,5,6,7], Mercedes Marín[5,6], Álvaro Pedraz[8], Mª Carmen Fariñas[9,10,11], Valentín Tascón[12], Arístides de Alarcón[13,14,15], Raquel Rodríguez-García[16,17], José María Miró[18,19], Josune Goikoetxea[20], Guillermo Ojeda-Burgos[21], Francesc Escrihuela-Vidal[19,22,23], Jorge Calderón-Parra[1,2], On behalf of the GAMES investigators[¶]

1 Unit of Infectious Diseases, Department of Internal Medicine, University Hospital Puerta de Hierro, Majadahonda, Spain, 2 Instituto Investigación Sanitaria Puerta de Hierro—Segovia de Arana (IDIPHSA), Majadahonda, Spain, 3 Autonomous University of Madrid, Majadahonda, Spain, 4 Department of Cardiology, University Hospital Puerta de Hierro, Majadahonda, Spain, 5 Department of Clinical Microbiology and Infectious Diseases, University General Hospital Gregorio Marañón, Madrid, Spain, 6 CIBER Enfermedades Respiratorias-CIBERES (CB06/06/0058), Madrid, Spain, 7 Complutense University of Madrid, Madrid, Spain, 8 Department of Cardiac Surgery, University General Hospital Gregorio Marañón, Madrid, Spain, 9 Department of Infectious Diseases, University Hospital Marqués de Valdecilla-IDIVAL, Santander, Spain, 10 CIBER de Enfermedades Infecciosas-CIBERINFEC (CB21/13/00068), Institute of Health Carlos III, Madrid, Spain, 11 University of Cantabria, Santander, Spain, 12 Department of Cardiovascular Surgery, University Hospital Marqués de Valdecilla, Santander, Spain, 13 Clinical Unit of Infectious Diseases, Microbiology and Preventive Medicine, Infectious Diseases Research Group Institute of Biomedicine of Seville (IBiS), Seville, Spain, 14 University of Seville/CSIC/University, Seville, Spain, 15 Hospital Virgen del Rocío, Seville, Spain, 16 Department of Intensive Medicine, University Hospital Central of Asturias, Oviedo, Spain, 17 University of Oviedo, Oviedo, Spain, 18 Infectious Diseases Service, Hospital Clinic-IDIBAPS, Barcelona, Spain, 19 University of Barcelona, Barcelona, Spain, 20 Department of Infectious Diseases, University Hospital de Cruces, Bilbao, Spain, 21 Department of Internal Medicine, University Hospital Virgen de la Victoria, Málaga, Spain, 22 Department of Infectious Diseases, University Hospital of Bellvitge, Barcelona, Spain, 23 Research Institut of Biomedicine of Bellvitge, Barcelona, Spain

¶ GAMES investigators are provided in Appendix (S1 Data).
* aramos220@gmail.com

**Data Availability Statement:** There is a restriction when it comes to sharing the data set, since both the Research Ethics Committee that approved the

## Abstract

### Background

Prosthetic valve endocarditis (PVE) is a serious infection associated with high mortality that often requires surgical treatment.

### Methods

Study on clinical characteristics and prognosis of a large contemporary prospective cohort of prosthetic valve endocarditis (PVE) that included patients diagnosed between January 2008 and December 2020. Univariate and multivariate analysis of factors associated with in-hospital mortality was performed.

study and the data protection legislation (Regulation (EU) 2016/679 of the European Parliament and of the European Council of 27 April 2016 on Data Protection (RGPD) only allow sharing patient data with health authorities and/or third parties if there is express consent from the data subject. The informed consent signed by our patients did not include the possibility that third parties could freely access their medical information containing some particularly sensitive data such as date of birth, initials, date of admission and discharge and the hospital where the patient was admitted. However, this information can be obtained in justified cases by contacting Ivan Adan from the technical office of the GAMES research network by e-mail: games08@gmail.com.

**Funding:** The author received no specific funding for this work.

**Competing interests:** : Dr. Ojeda-Burgos has received grants for assistance to medical meetings from Pfizer, Merck Sharp & Dohme, Gilead, Janssen, and Angelini; and has been paid as a speaker in medical meetings from Janssen, Gilead, and Merck Sharp & Dohme. Dr. Miró has received consulting honoraria and/or research grants from Angelini, Bristol-Myers Squibb, Contrafect, Genentech, Gilead Sciences, Merck Sharp and Dohme, Medtronic, Novartis, Pfizer, and ViiV. All authors (including Dr. Ojeda-Burgos and Dr. Miró) have reported that they have no interest conflicts to disclose related to the contents of this paper. This does not alter our adherence to PLOS ONE policies on sharing data and materials.

**Abbreviations:** IE, infective endocarditis; PVE, prosthetic valve endocarditis; NVE, native valve endocarditis; CoNS, coagulase-negative staphylococci; CT, Computed tomography.

## Results

The study included 1354 cases of PVE. The median age was 71 years with an interquartile range of 62–77 years and 66.9% of the cases were male. Patients diagnosed during the first year after valve implantation (early onset) were characterized by a higher proportion of cases due to coagulase-negative staphylococci and *Candida* and more perivalvular complications than patients detected after the first year (late onset). In-hospital mortality of PVE in this series was 32.6%; specifically, it was 35.4% in the period 2008–2013 and 29.9% in 2014–2020 (p = 0.031). Variables associated with in-hospital mortality were: Age-adjusted Charlson comorbidity index (OR: 1.15, 95% CI: 1.08–1.23), intracardiac abscess (OR:1.78, 95% CI:1.30–2.44), acute heart failure related to PVE (OR: 3. 11, 95% CI: 2.31–4.19), acute renal failure (OR: 3.11, 95% CI:1.14–2.09), septic shock (OR: 5.56, 95% CI:3.55–8.71), persistent bacteremia (OR: 1.85, 95% CI: 1.21–2.83) and surgery indicated but not performed (OR: 2.08, 95% CI: 1.49–2.89). In-hospital mortality in patients with surgical indication according to guidelines was 31.3% in operated patients and 51.3% in non-operated patients (p<0.001). In the latter group, there were more cases of advanced age, comorbidity, hospital acquired PVE, PVE due to *Staphylococcus aureus*, septic shock, and stroke.

## Conclusions

Not performing cardiac surgery in patients with PVE and surgical indication, according to guidelines, has a significant negative effect on in-hospital mortality. Strategies to better discriminate patients who can benefit most from surgery would be desirable.

## Introduction

Prosthetic valve endocarditis (PVE) constitutes 20–30% of cases of infective endocarditis (IE) and is associated with high mortality [1, 2]. The lesser detection of signs of PVE on imaging techniques, such as vegetations and/or periannular complications typical of IE, and the possible visualization of residual findings using these techniques, which could be explained by the previous valve surgery itself, makes it more challenging to establish an adequate diagnosis in cases of prosthetic valve endocarditis (PVE) compared to native valve endocarditis (NVE) [1–4].

The frequent extension of the infection around the prosthetic valve implies a greater challenge in surgical treatment compared to NVE [3, 5–7]. Patients who present surgical indication but do not undergo surgery are a matter of great concern that should be carefully analyzed for its prognostic implications [7, 8]. The percentage of patients with PVE and surgical indication who ultimately do not undergo surgery was higher than 40% in some series [9]. Among the reasons given for discouraging surgical intervention in these patients are severe sepsis, cerebral embolism, cardiogenic shock, and acute renal failure [10]. Improving knowledge of the prognostic variables of patients with PVE and the causes of disregard for surgical treatment seem to be important aspects to optimize the clinical management of these patients [11–13].

The aim of this study was to describe the clinical presentation and prognosis of patients with PVE. Specifically, we sought to explore the clinical characteristics and the prognosis of patients with surgical indication according to the guidelines who did not undergo surgery. To achieve this objective, we conducted an analysis of patients included in a large contemporary cohort of IE cases.

## Patients and methods

From January 2008 to December 2020, consecutive patients with a definite diagnosis IE, according to Duke's modified criteria, were prospectively included. These patients received treatment in a group of Spanish hospitals, collectively serving approximately 30% of the nation's population. At each center, a multidisciplinary team completes a standardized form with the IE episode and a follow-up form after one year of the episode. The register included sections for demographic, clinical, microbiological, echocardiographic, management and prognostic information. The cohort registration received approval of regional and local ethics committees. Specifically, the Ethics and Clinical Research Board of one of participant hospitals approved the study protocol and publication of data (Gregorio Marañón Hospital in Madrid, number 18/07). Informed consent was obtained in cases where the patient could be adequately informed. For patients in coma or incapable of giving consent, the ethics committees waived the requirement for investigators to obtain consent to avoid patient inclusion bias. Data and samples were collected from January 2008 to December 2021. Subsequently, the study data were analyzed during the years 2022 and 2023. The authors did not have access to information that could identify individual participants during or after data collection.

## Definitions

**General variables.** General definitions correspond to those published in other studies on endocarditis [14, 15]. Healthcare-associated infections were defined as previously published [16]. Patients were categorized into either early or late PVE, depending on whether the diagnosis was made before or after the first year following prosthetic valve implantation, respectively [1, 12]. Persistent bacteremia was defined as persistence of positive blood cultures after 7 days of appropriate antibiotic treatment initiation. Systemic embolization included embolism to any major arterial vessel, excluding stroke, which was defined by acute neurological deficit of vascular origin lasting >24 hours. Episodes with neurological symptoms lasting less than 24 hours, but showing imaging scans suggestive of infarction, were classified as stroke [17].

**Exposures of interest.** Surgical indications followed the latest current European guidelines available at the time of diagnosis [2, 18, 19]. Particular focus was directed to identifying patients with surgical indications and, within this group, those who were not operated on.

**Outcomes of interest.** In-hospital mortality and 1-year mortality were defined as death from any cause during hospital admission or within the 365 days following admission in which PVE was treated, respectively. Recurrent IE was defined as a new episode of IE during the first year of follow-up [20].

## Patients

The study analyzed demographic, clinical, echocardiographic, and treatment data of the included patients, as well as morbidity and mortality both at admission and during the first year of follow-up. Endocarditis on transcatheter aortic valve replacement and infection of non-valve aortic graft were not included in the study due to their distinctive clinical characteristics [21, 22]. Patients with atrial or ventricular septal defect closure or cardiovascular implantable electronic devices infection were included only if they had a concomitantly infected prosthetic valve.

## Statistical analysis

Categoric variables are expressed as absolute numbers and percentages. Quantitative variables are expressed as median and interquartile range (IQR). Categorical variables were compared

using $\chi^2$ test or Fisher test when necessary. Quantitative variables were compared using Mann-Whitney's U. In the comparison of risk factors for mortality, those variables with $p < 0.10$ in univariant analysis and that were considered clinically significant, were included in a multivariate logistic regression model, with a maximum of one variable for every 10 events (deaths). The goodness of fit of the final multivariate mode was assessed again by the Hosmer-Lemeshow test. Adjusted odds ratios and its 95% confident interval are provided. Bilateral p-value below 0.05 was considered statistically significant. All statistical analyses were performed with SPSS version 25 software (SPSS INC., Chicago, Illinois, USA). The data on which this study is based are available upon reasonable request through the technical office of the research network [(Spanish collaboration on endocarditis (GAMES)] which can be contacted via this e-mail: games08@gmail.com.

## Results

During the study period, a total of 4454 consecutive cases with definitive IE were identified. Among them, 1354 cases (30.4%) corresponded to PVE (Fig 1). Out of the PVE cases, 492 (36.3%) were diagnosed within the first year after prosthetic valve implantation (early PVE) while 862 cases (63.6%) were diagnosed after the first year (late PVE). The proportion of PVE cases over the total of IE cases was 29.7% between 2008 and 2013 (672 out of 2264 cases) and 31.4% between 2014 and 2020 [(682 out of 2190 cases); p = 0.290]). Among the PVE cases, 633 involved mechanical valves (47%), and 718 involved biological valves (53.9%). The number of infected mechanical prostheses in the mitral position was 354 out of 515 prosthetic valves (68,7%) and 358 out of 969 prosthetic valves in the aortic position (36.9%; p<0.001). Simultaneous involvement of prosthetic valves in both the aortic and mitral positions occurred in 173 cases (12.7%).

### Clinical characteristics and outcome of patients with PVE

Patients with PVE were older, had a higher comorbidity burden, a greater proportion of patients with surgical indications who did not undergo surgery, and higher mortality compared to patients with NVE (Table 1). The indication of surgery of the episodes of PVE compared to episodes of NVE are shown in Table 1S in the S1 Data. Table 2S in the S1 Data

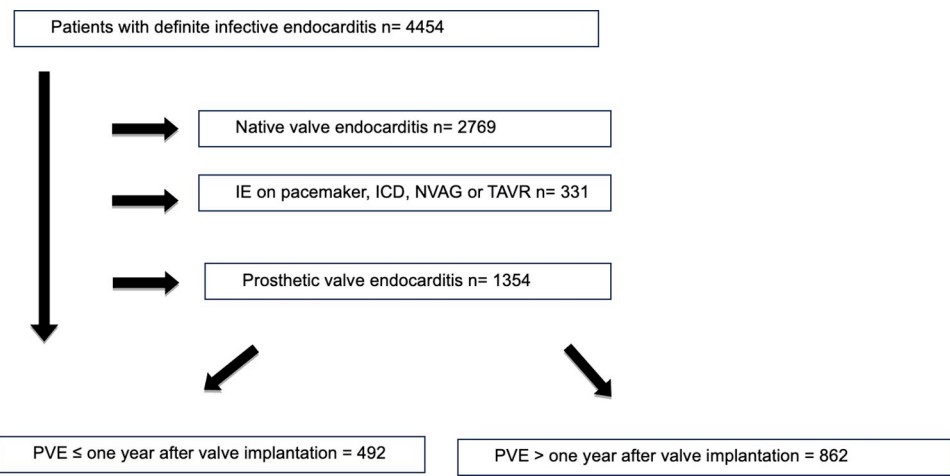

**Fig 1. Flowchart of patients presenting with definite or possible infective endocarditis (IE) according to the type of affected valve (games cohort 2008–2020).** ICD: implantable cardioverter defibrillator. NVAG: non-valve aortic graft. TAVR: transcatheter aortic valve replacement. PVE: prosthetic valve endocarditis.

**Table 1. Characteristics of patients with native valve endocarditis compared patients with prosthetic valve endocarditis.**

| | Native (n = 2769) | Prosthetic (n = 1354) | Overall (N = 4123) | p-value |
|---|---|---|---|---|
| Age. years (IQR) | 66 (53–76) | 71 (62–77) | 68 (57–76) | <0.001 |
| Male gender | 1897 (68.5) | 903 (66.9) | 2800 (67.9) | 0.240 |
| Hospital-acquired | 590 (21.3) | 515 (38.0) | 1105 (26.8) | <0.001 |
| Site of infection | | | | |
| Aortic | 1386 (50.1) | 969 (71.6) | 2355 (57.1) | <0.001 |
| Mitral | 1469 (53.1) | 515 (38.0) | 1984 (48.1) | <0.001 |
| Comorbidity | | | | |
| Coronary disease | 556 (20.1) | 471 (34.7) | 1027 (24.9) | <0.001 |
| Chronic heart failure | 661 (23.9) | 609 (44.9) | 1270 (30.8) | <0.001 |
| Intravenous drug user | 109 (3.9) | 10 (0.7) | 119 (2.8) | <0.001 |
| Cerebrovascular disease | 282 (10.2) | 235 (17.3) | 517 (12.5) | <0.001 |
| Chronic renal failure | 664 (24.0) | 357 (26.4) | 1021 (24.7) | 0.095 |
| Chronic liver disease | 339 (12.2) | 95 (7.1) | 434 (10.5) | <0.001 |
| Age-adjusted Charlson index (IQR) | 4 (3–6) | 5 (3–7) | 5 (3–7) | <0.001 |
| Microbiology | | | | |
| Gram-positive bacteria | | | | |
| *Staphylococcus aureus* | 766 (27.7) | 208 (15.4) | 974 (23.6) | <0.001 |
| MRSA | 116 (4.1) | 42 (3.1) | 158 (3.8) | 0.088 |
| Coagulase-negative staphylococci | 285 (10.3) | 437 (32.3) | 722 (17.5) | <0.001 |
| *Enterococcus spp* | 441 (15.9) | 217 (16.0) | 658 (15.9) | 0.934 |
| Streptococcus spp | 932 (33.7) | 262 (19.4) | 1194 (28.9) | <0.001 |
| Gram-negative bacilli | 91 (3.3) | 63 (4.7) | 154 (3.7) | 0.030 |
| Anaerobic bacteria | 16 (0.6) | 30 (2.2) | 46 (1.1) | <0.001 |
| Fungi | | | | |
| *Candida spp* | 31 (1.1) | 33 (2.4) | 64 (1.5) | 0.001 |
| Other fungi | 10 (0.4) | 2 (0.1) | 12 (0.3) | 0.358 |
| Polymicrobial | 34 (1.2) | 18 (1.3) | 52 (1.2) | 0.784 |
| Other microorganisms | 65 (2.3) | 36 (2.7) | 101 (2.4) | 0.544 |
| Negative cultures (no growth) | 78 (2.8) | 37 (2.7) | 115 (2.7) | 0.877 |
| Echocardiographic findings | | | | |
| Vegetation | 2359 (85.2) | 928 (68.5) | 3287 (79.7) | <0.001 |
| Intracardiac complications | 971 (35.1) | 569 (42.0) | 1540 (37.3) | <0.001 |
| Valve perforation or rupture | 633 (22.9) | 50 (3.6) | 683 (16.5) | <0.001 |
| Pseudoaneurysm | 144 (5.2) | 148 (10.9) | 292 (7.0) | <0.001 |
| Perivalvular abscess | 366 (13.2) | 460 (34.0) | 826 (20.0) | <0.001 |
| Intracardiac fistula | 59 (2.1) | 63 (4.6) | 122 (2.9) | <0.001 |
| Clinical course | | | | |
| Acute heart failure | 1271 (45.9) | 542 (40.0) | 1813 (43.9) | <0.001 |
| Persistent bacteremia | 326 (11.8) | 153 (11.3) | 479 (11.6) | 0.656 |
| Stroke | 602 (21.7) | 320 (23.6) | 922 (22.3) | 0.171 |
| Embolism [a] | 721 (26.0) | 285 (21.0) | 1006 (24.3) | <0.001 |
| Mycotic aneurism | 75 (2.7) | 30 (2.2) | 105 (2.5) | 0.345 |
| Acute renal failure | 948 (34.2) | 571 (42.1) | 1519 (36.8) | <0.001 |
| Septic shock | 376 (13.6) | 183 (13.5) | 559 (13.5) | 0.955 |
| Surgical indication | 1887 (68.1) | 1009 (74.5) | 2896 (70.2) | <0.001 |
| Surgery performed [b] | 1306 (69.2) | 650 (64.4) | 1956 (67.5) | 0.009 |
| Surgery indicated. not performed | 581 (30.8) | 359 (35.6) | 940 (32.4) | <0.001 |

*(Continued)*

**Table 1.** (Continued)

|  | Native (n = 2769) | Prosthetic (n = 1354) | Overall (N = 4123) | p-value |
|---|---|---|---|---|
| In-hospital mortality | 709 (25.6) | 442 (32.6) | 1151 (27.9) | <0.001 |
| First year mortality | 876 (31.6) | 507 (37.4) | 1383 (33.5) | <0.001 |
| Recurrence [c] | 28 (1.3) | 21 (2.3) | 49 (1.6) | 0.063 |

IQR: Interquartile range. MRSA: methicillin-resistant *S. aureus*.

[a] Excluding cases with stroke.

[b] Percentages calculated considering only patients with surgical indications.

[c] during the first year after diagnosis calculated on patients discharged from the hospital (n = 2972).

Seventy cases (5.1%) showed concomitant involvement of native and prosthetic valves.

presents a comparison of clinical characteristics between patients with only aortic or mitral prosthetic valve involvement. Patients with aortic PVE were older, had a higher incidence of early PVE, a higher frequency of coagulase-negative staphylococci (CoNS), and more intracardiac complications than patients with mitral PVE.

CoNS were the most common bacteria causing PVE in this series. The proportion of CoNS PVE cases increased from 30.5% in the first period (2008–2013) to 34% in the second period (2014–2020), although this difference was not statistically significant (p = 0.167, Table 3S in the S1 Data). In addition, CoNS were identified in 43.5% of early PVE cases during the first period (2008–2013) and in 54.3% during the second period (2014–2020; p = 0.017). *Staphylococcus aureus* caused 20% of PVE cases on mechanical valves and 11.3% on biological valves (p<0.0019. in cases due to CoNS, this proportion was 27.8% and 36.2%, respectively (p = 0.001).

In-hospital mortality of PVE in this series was 32.6% (Table 1). Table 2 shows the characteristics of the patients according to in-hospital mortality. Variables independently associated with in-hospital mortality were age-adjusted Charlson comorbidity index (OR: 1.15, 95% CI: 1.08–1.23), intracardiac abscess (OR:1.78, 95% CI:1.30–2.44), acute heart failure related to PVE (OR: 3. 11, 95% CI: 2.31–4.19), acute renal failure (OR: 3.11, 95% CI:1.14–2.09), septic shock (OR: 5.56, 95% CI:3.55–8.71), persistent bacteremia (OR: 1.85, 95% CI: 1.21–2.83) and surgery indicated but not performed (OR: 2.08, 95% CI: 1.49–2.89) (Table 3). Given the significant association between septic shock and in-hospital mortality, a multivariate analysis was performed without this variable (Table 4). The result was very similar, except that mitral involvement and PVE due to *S. aureus* were identified as independent prognostic variables in this second analysis, and persistent bacteremia was no longer statistically significant.

A comparison of patient characteristics was made according to the period in which the diagnosis was made (2008–2013 vs. 2014–2020; Table 3S in the S1 Data). It was evident that comorbidity, late PVE, intracardiac complications and septic shock were more frequent in patients treated during the second period (2014–2020). Additionally, in-hospital mortality in the second period (29.9%) was lower than in the first period (35.4%, p = 0.031).

## Clinical characteristics and outcome of patients with surgical indication that were not operated on

One thousand and nine patients presented surgical indication (74.5%). Six hundred fifty patients (64.4%) underwent surgery, and 359 patients (35.6%) were managed conservatively, with antibiotic treatment only (Table 5). Patients who did not undergo surgery were older, had higher frequency of chronic lung disease, chronic heart failure, peripheral vascular disease, neoplasia, previous renal failure and chronic liver disease with a significant difference in the

**Table 2. Characteristics of patients with PVE according to in-hospital mortality.**

| | Survivors (n = 912) | Non-survivors (n = 442) | p-value |
|---|---|---|---|
| Age. years (IQR) | 69 (61–76) | 73 (65–78) | <0.001 |
| Male gender | 623 (68.3) | 280 (63.3) | 0.069 |
| Hospital-acquired | 313 (34.3) | 202 (45.7) | <0.001 |
| Site of infection | | | |
| Aortic | 645 (70.7) | 324 (73.3) | 0.324 |
| Mitral | 322 (35.3) | 193 (43.7) | 0.003 |
| Tricuspid | 13 (1.4) | 3 (0.7) | 0.293 |
| Pulmonary | 25 (2.7) | 2 (0.5) | 0.005 |
| Comorbidity | | | |
| Chronic heart failure | 380 (41.6) | 229 (51.8) | <0.001 |
| Diabetes mellitus | 250 (27.4) | 155 (35.0) | 0.004 |
| Intravenous drug user | 10 (1.0) | 0 | - |
| Peripheral vascular disease | 70 (7.6) | 57 (12.8) | 0.002 |
| Cerebrovascular disease | 153 (16.6) | 82 (18.5) | 0.419 |
| Neoplasia | 139 (15.2) | 77 (17.4) | 0.304 |
| Chronic renal failure | 207 (22.7) | 150 (33.9) | <0.001 |
| Chronic liver disease | 55 (6.0) | 40 (9.0) | 0.041 |
| Congenital heart disease | 68 (7.4) | 16 (3.6) | 0.006 |
| Age-adjusted Charlson index (IQR) | 4 (3–6) | 5 (4–7) | <0.001 |
| Early PVE | 319 (35.0) | 173 (39.1) | 0.135 |
| Late PVE | 593 (65.0) | 269 (60.9) | 0.135 |
| Microbiology | | | |
| Gram-positive bacteria | | | |
| Staphylococcus aureus | 105 (11.5) | 103 (23.3) | <0.001 |
| CoNS | 285 (31.3) | 152 (34.4) | 0.247 |
| Enterococcus | 161 (17.7) | 56 (12.7) | 0.019 |
| Streptococcus | 199 (21.8) | 63 (14.3) | 0.001 |
| Gram-negative bacilli | 45 (4.9) | 18 (4.1) | 0.480 |
| Anaerobic bacteria | 24 (2.6) | 6 (1.4) | 0.135 |
| Fungi | | | |
| Candida | 17 (1.9) | 16 (3.6) | 0.049 |
| Other fungal species | 1 (0.1) | 1 (0.2) | 0.546 |
| Polymicrobial | 11 (1.2) | 7 (1.6) | 0.569 |
| Other microorganisms | 27 (3.0) | 9 (2.0) | 0.322 |
| Echocardiographic findings | | | |
| Vegetation | 615 (67.4) | 313 (70.8) | 0.209 |
| Intracardiac complications | 344 (37.7) | 225 (50.9) | <0.001 |
| Valve perforation or rupture | 25 (2.7) | 25 (5.6) | 0.008 |
| Pseudoaneurysm | 90 (9.8) | 58 (13.1) | 0.072 |
| Perivalvular abscess | 286 (31.4) | 174 (39.4) | 0.012 |
| Intracardiac fistula | 40 (4.3) | 23 (5.5) | 0.503 |
| Clinical course | | | |
| Acute heart failure | 271 (29.7) | 271 (61.3) | <0.001 |
| Persistent bacteremia | 86 (9.4) | 67 (15.1) | 0.002 |
| Stroke | 176 (19.2) | 144 (32.5) | <0.001 |
| Embolism [a] | 192 (21.0) | 93 (21.0) | 0.996 |
| Acute renal failure | 317 (34.7) | 254 (57.4) | <0.001 |

(*Continued*)

**Table 2.** (Continued)

|  | Survivors (n = 912) | Non-survivors (n = 442) | p-value |
|---|---|---|---|
| Septic shock | 43 (4.7) | 140 (31.6) | <0.001 |
| Surgical indication | 622 (68.2) | 387 (87.6) | <0.001 |
| Surgery performed [b] | 447 (71.9) | 203 (52.4) | 0.238 |
| Surgery indicated not performed | 175 (28.1) | 184 (47.6) | <0.001 |

IQR: Interquartile range.

[a] Excluding cases with stroke.

[b] Percentages calculated considering only patients with surgical indications.

age-adjusted Charlson index 6 points (IQR: 4–8 points) versus 4 points (IQR: 3–6 points; p = <0.001), respectively. Nosocomial acquisition of infection, PVE due to *S. aureus*, septic shock and brain involvement were also more frequent among the non-operated patients (Table 2). In-hospital mortality was significantly higher among patients who did not undergo surgery (51.3%) compared to those who underwent surgery (31.3%, p<0.001). Fig 2 shows the survival during the first year in patients without surgical indication, with surgical indication who underwent surgery and with surgical indication who did not undergo surgery.

The reasons given for not performing the intervention were as follows: severe hemodynamic instability leading to poor prognosis (75 patients, 20.9%), neurological complications (73 patients, 20.3%), challenging surgical procedures (45 patients, 12.5%), other medical causes (74 patients, 20.6%), patient refusal (52 patients, 14.5%) and death of the patient during the discussion of the feasibility of intervention (40 patients, 11.1%).

## Clinical characteristics and outcome of patients with PVE according to time of onset

Patients diagnosed during the first year after prosthetic valve implantation had a higher frequency of coronary artery disease, chronic renal failure or liver disease, hospital acquisition, aortic PVE and intracardiac complications (such as pseudoaneurysm or abscess) and a lower frequency of mitral and tricuspid valve involvement compared to patients diagnosed with late PVE (Table 4S in the S1 Data). Regarding microbiology, there were more cases due to CoNS and *Candida* and fewer cases of *S. aureus* and *Streptococcus*. Mortality in patients with early PVE was 35.2% and that of patients with late PVE was 31.2% (p = 0.132, Table 4S in the S1

**Table 3. Multivariate analysis of clinical factors of PVE associated with in-hospital mortality.**

|  | OR | CI 95% | p-value |
|---|---|---|---|
| Age, years | 1.08 | 0.99–1.22 | 0.255 |
| Mitral affected | 1.33 | 0.97–1.81 | 0.070 |
| Age-adjusted Charlson Comorbidity, points | 1.15 | 1.08–1.23 | <0.001 |
| Staphylococcus aureus | 1.38 | 0.91–2.09 | 0.120 |
| Acute heart failure | 3.11 | 2.31–4.19 | <0.001 |
| Persistent bacteremia | 1.85 | 1.21–2.83 | 0.005 |
| Septic Shock | 5.56 | 3.55–8.71 | <0.001 |
| Acute renal failure | 1.55 | 1.14–2.09 | 0.005 |
| Nosocomial | 1.23 | 0.91–1.67 | 0.165 |
| Intracardiac Abscess | 1.78 | 1.30–2.44 | <0.001 |
| Surgery indicated. not performed | 2.08 | 1.49–2.89 | <0.001 |

**Table 4. Multivariate analysis of clinical factors of PVE associated with in-hospital mortality without considering "septic shock".**

|  | OR | CI 95% | p-value |
|---|---|---|---|
| Age, years | 1.07 | .99–1.01 | 0.226 |
| Mitral affected | 1.36 | 1.03–1.78 | 0.026 |
| Age-adjusted Charlson Comorbidity, points | 1.12 | 1.05–1.19 | <0.001 |
| Staphylococcus aureus | 1.86 | 1.31–2.64 | <0.001 |
| Acute heart failure | 3.08 | 2.37–4.01 | <0.001 |
| Persistent bacteremia | 1.46 | .99–1.01 | 0.055 |
| Acute renal failure | 1.82 | 1.39–2.37 | <0.01 |
| Nosocomial | 1.35 | 1.03–1.76 | 0.028 |
| Intracardiac Abscess | 1.66 | 1.26–2.20 | <0.001 |
| Surgery indicated. not performed | 2.34 | 1.75–3.11 | <0.001 |

Data) Comparison of the characteristics of patients with PVE who had surgical indication depending on whether they underwent surgery or not and considering separately by the time of onset of PVE (early or late) is presented in Tables 5S and 6S in the S1 Data.

## Discussion

We present a comprehensive series of PVE characterized by patients with advanced age and marked comorbidity, as well as by an important role played by CoNS and by the fact that one third of the cases were not operated despite having a surgical indication.

### Clinical characteristics and outcome of patients with PVE

Patients with PVE are generally older and have more comorbidities compared to patients with NVE, as has also been evidenced in previous studies [1, 3, 23]. A higher frequency of cases due to CoNS with less involvement of *S. aureus* and *Streptococcus* has also been reported [3, 7]. The incidence of PVE due to CoNS was also higher in our study than the 16.9% recorded in another large series of patients diagnosed between 2000 and 2005 [13]. As an additional fact about etiology, it should be noted the greater tendency for *S. aureus* to infect mechanical valves and for CoNS and *Enterococcus* to infect biological valves. Although we have not found other studies with similar results, we consider it relevant to study in the future the possible differences in the adherence of bacteria depending on the material of which prosthetic valves are made, due to their possible preventive or therapeutic implications.

Mortality among PVE cases was higher than in NVE cases, however, however, there are studies showing that mortality, when adjusted for risk factors, may be equal to or even lower than that of patients with NVE [3]. Our study identified several variables independently associated with in-hospital mortality, consistent with previous research. These included baseline patient characteristics (age and comorbidity) [13, 23], the development of acute heart failure [2, 7, 13, 24], perivalvular complications [7, 8, 11], severity of infection indicated by septic shock or persistent bacteremia [13] and cases with surgical indication that were not operated on [8, 25]. The reduction in mortality over time observed in this study has also been evidenced in previous investigations [23, 26]. However, we have not found a clear reason for this finding beyond the slightly higher number of patients who underwent TEE during the second period compared to the first. Advances made in recent years in diagnostic acuity, imaging techniques and surgical treatment may have influenced the reduction in mortality during the second period despite including patients with higher severity [8, 23].

**Table 5. Characteristics of patients with PVE and surgical indication according to whether the patient underwent surgery.**

| | Surgery performed (n = 650) | Surgery not performed (n = 359) | p-value |
|---|---|---|---|
| Age. years (IQR) | 69 (59–75) | 73 (65–79) | <0.001 |
| Male gender | 449 (69.0) | 233 (64.9) | 0.175 |
| Hospital-acquired | 247 (38.0) | 158 (44.0) | 0.062 |
| Site of infection | | | |
| Aortic | 477 (73.4) | 264 (73.5) | 0.958 |
| Mitral | 229 (35.2) | 147 (40.9) | 0.072 |
| Tricuspid | 7 (1.1) | 8 (2.2) | 0.148 |
| Pulmonary | 11 (1.7) | 9 (2.5) | 0.374 |
| Comorbidity | | | |
| Chronic heart failure | 273 (42.0) | 195 (54.3) | 0.001 |
| Diabetes mellitus | 186 (28.6) | 107 (29.8) | 0.690 |
| Intravenous drug user | 5 (0.8) | 1 (0.3) | 0.571 |
| Peripheral vascular disease | 49 (7.5) | 42 (11.7) | 0.027 |
| Cerebrovascular disease | 111 (17.0) | 60 (16.7) | 0.883 |
| Neoplasia | 74 (11.3) | 75 (20.8) | <0.001 |
| Chronic renal failure | 137 (21.1) | 125 (34.8) | <0.001 |
| Chronic liver disease | 35 (5.3) | 42 (11.7) | <0.001 |
| Congenital heart disease | 53 (8.1) | 18 (5.0) | 0.062 |
| Age-adjusted Charlson index (IQR) | 4 (3–6) | 6 (4–8) | <0.001 |
| Early PVE | 255 (39.2) | 134 (37.3) | 0.552 |
| Late PVE | 395 (60.8) | 225 (62.7) | 0.552 |
| Microbiology | | | |
| Gram-positive bacteria | | | |
| *Staphylococcus aureus* | 86 (13.2) | 81 (22.6) | <0.001 |
| CoNS | 245 (37.7) | 120 (33.4) | 0.177 |
| *Enterococcus* | 88 (13.5) | 51 (14.2) | 0.768 |
| *Streptococcus* | 108 (16.6) | 57 (15.9) | 0.762 |
| Gram-negative bacilli | 20 (3.1) | 18 (5.0) | 0.122 |
| Anaerobic bacteria | 25 (3.8) | 0 | - |
| Fungi | | | |
| *Candida* | 18 (2.8) | 11 (3.1) | 0.788 |
| Other fungal species | 1 (0.2) | 1 (0.3) | 0.670 |
| Polymicrobial | 8 (1.2) | 5 (1.4) | 0.827 |
| Other microorganisms | 23 (3.5) | 6 (1.7) | 0.089 |
| Echocardiographic findings | | | |
| Vegetation | 460 (70.8) | 251 (69.9) | 0.776 |
| Intracardiac complications | 373 (57.4) | 158 (44.0) | <0.001 |
| Valve perforation or rupture | 35 (5.3) | 14 (3.9) | 0.293 |
| Pseudoaneurysm | 91 (14.0) | 45 (12.5) | 0.514 |
| Perivalvular abscess | 311 (47.8) | 120 (33.4) | <0.001 |
| Intracardiac fistula | 43 (6.6) | 17 (4.7) | 0.227 |
| Clinical course | | | |
| Acute heart failure | 294 (45.2) | 177 (49.3) | 0.214 |
| Persistent bacteremia | 67 (10.3) | 51 (14.2) | 0.065 |
| Stroke | 146 (22.4) | 102 (28.4) | 0.036 |
| Embolism [a] | 146 (22.4) | 82 (22.8) | 0.89 |
| Acute renal failure | 289 (44.4) | 169 (47.0) | 0.425 |

*(Continued)*

**Table 5.** (Continued)

| | Surgery performed (n = 650) | Surgery not performed (n = 359) | p-value |
|---|---|---|---|
| Septic shock | 74 (11.3) | 86 (23.9) | <0.001 |
| In-hospital mortality | 203 (31.3) | 184 (51.3) | <0.001 |
| First year mortality | 232 (35.7) | 201 (55.9) | <0.001 |
| Recurrence | 7 (1.5) | 4 (2.8) | 0.540 |

IQR: Interquartile range. CoNS: Coagulase-negative staphylococci.

[a] Excluding cases with stroke

## Clinical characteristics and outcome of patients with surgical indication who were not operated on

As seen in previous studies, the decision to forgo surgery in patients with surgical indication has a significant impact on prognosis [8]. Among patients who did not undergo surgery, there was a notable tendency to be older and with more comorbidities. Although patients older than 65 years tend to have worse prognosis due to comorbidities, age alone should not be an exclusive factor to exclude surgery [27–29]. Of note, patients with chronic liver disease underwent surgery less frequently and experienced higher mortality. It is suggested to consider the status of liver disease (Child-Pugh score) before ruling out surgical intervention in these cases [28]. Surprisingly, cases due to *S. aureus*, which usually require surgical treatment, were operated less frequently. This could be explained by the higher frequency of severe systemic infection, secondary septic foci, or greater surgical complexity in these patients [9, 29]. Similarly, cases with central nervous system involvement were also less likely to receive surgery. Adequate assessment of the type and extent of stroke (ischemic or hemorrhagic) is essential before discouraging surgery [30]. Considering the improved survival rates in recent years, physicians

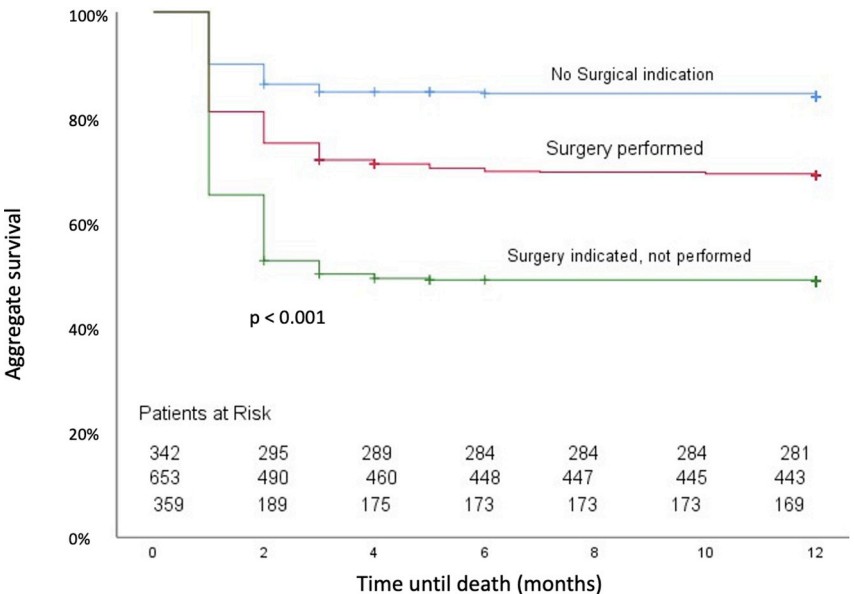

Global comparisons. Log Rank /Mantel-Cox χ²: 121.1; p < 0.001. Test for equality of survival distribution for different levels of cases. PVE: Prosthetic valve endocarditis

**Fig 2. Survival of patient with PVE according to surgery performance.**

should strive to identify patients with poor prognostic factors who may still benefit from surgery [1, 27, 31]. Strategies to reduce the number of patients denied surgery may include better patient education about treatment options, adherence to recommended surgical timelines (emergent, urgent, or elective), and facilitation of transfers to hospitals with expertise in complex surgery.

### Clinical characteristics of patients with PVE according to time of onset

When comparing patients who were diagnosed within the first year after valve implantation with those diagnosed later, we observed more cases of nosocomial origin, as would be expected. A higher incidence of intracardiac complications during the first year was also detected, emphasizing the increased importance, if possible, of performing transesophageal echocardiography and other imaging tests such as positron emission tomography/computed tomography (PET/CT) or cardiac CT in suspected cases of early PVE [8, 11, 32]. The percentage of CoNS causing late PVE was lower than that observed in early PVE cases. Despite this fact, empirical coverage for CoNS could be advisable in late PVE given that it originated 23% of these cases. The occurrence of PVE due to *Candida* was also significantly lower in late cases (1.2% vs. 4.7% in early cases). Given these low figures, empirical treatment with antifungals in early PVE may not be justified.

### Limitations

Firstly, we must acknowledge the extended duration of the study, which could have led to differences in the diagnosis and treatment approaches over time. It is also necessary to take into account the impact of changes in diagnosis and treatment of the different IE guidelines considered over time on the homogeneity of the patients included in the study. Finally, we must point out the fact that many patients were referred from hospitals without cardiac surgery, which could have influenced the etiology and certain characteristics of the patients studied. More severe or milder cases could have been transferred less frequently because surgical intervention can be ruled out at the outset. However, these differences should not be very important considering the fluid communication and adequate coordination between the hospitals without cardiac surgery and the referral hospitals.

## Conclusions

Patients with PVE account for nearly one third of all episodes of infective endocarditis (IE) and are characterized by advanced age, marked comorbidity, and a prominent role of CoNS, even in late-onset PVE. The proportion of patients with a surgical indication who do not undergo surgery is significant and is associated with higher mortality rates. Efforts should be made to better identify patients who might benefit most from surgery, including consideration of transfer to referral centers, in order to reduce rejection and delay in the performance of the surgery.

## Supporting information

**S1 Data.**
(DOCX)

## Acknowledgments

We thank Iván Adán for his task as data coordinator of the GAMES cohort and for his statistical support. We are grateful for the contribution of Juan Rivera Rodríguez in the grammatical revision of the manuscript.

## Author Contributions

**Conceptualization:** Antonio Ramos-Martínez, Fernando Domínguez, Patricia Muñoz, Mª Carmen Fariñas, José María Miró, Jorge Calderón-Parra.

**Data curation:** Antonio Ramos-Martínez, Fernando Domínguez, Patricia Muñoz, Álvaro Pedraz, Valentín Tascón, Arístides de Alarcón, Raquel Rodríguez-García, Josune Goikoetxea, Guillermo Ojeda-Burgos, Francesc Escrihuela-Vidal, Jorge Calderón-Parra.

**Formal analysis:** Antonio Ramos-Martínez, Valentín Tascón, Arístides de Alarcón, José María Miró, Josune Goikoetxea, Guillermo Ojeda-Burgos, Jorge Calderón-Parra.

**Investigation:** Antonio Ramos-Martínez, Mercedes Marín, Álvaro Pedraz, Mª Carmen Fariñas, Valentín Tascón, José María Miró.

**Methodology:** Antonio Ramos-Martínez, Patricia Muñoz, Mercedes Marín, Álvaro Pedraz, Mª Carmen Fariñas, Arístides de Alarcón, Raquel Rodríguez-García, José María Miró, Guillermo Ojeda-Burgos, Francesc Escrihuela-Vidal.

**Supervision:** Mercedes Marín, José María Miró, Josune Goikoetxea.

**Validation:** Josune Goikoetxea, Guillermo Ojeda-Burgos, Francesc Escrihuela-Vidal.

**Writing – original draft:** Antonio Ramos-Martínez, Fernando Domínguez, Jorge Calderón-Parra.

**Writing – review & editing:** Antonio Ramos-Martínez, Fernando Domínguez, Patricia Muñoz, Mercedes Marín, Álvaro Pedraz, Mª Carmen Fariñas, Valentín Tascón, Arístides de Alarcón, Raquel Rodríguez-García, José María Miró, Josune Goikoetxea, Guillermo Ojeda-Burgos, Francesc Escrihuela-Vidal, Jorge Calderón-Parra.

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
