## [Decision Letter · Decision Letter 0]

6 Jul 2023

PONE-D-23-16421Clinical presentation, microbiology, and prognostic factors of prosthetic valve endocarditis. Lessons learned from a large prospective registry.PLOS ONE

Dear Dr. Ramos-Martínez,

Thank you for submitting your manuscript to PLOS ONE. After careful consideration, we feel that it has merit but does not fully meet PLOS ONE’s publication criteria as it currently stands. Therefore, we invite you to submit a revised version of the manuscript that addresses the points raised during the review process.

We look forward to receiving your revised manuscript.

Kind regards,

Redoy Ranjan, MBBS, MRCSEd, Ch.M., MS (CV&TS), FACS

Academic Editor

PLOS ONE

Journal Requirements:

"Dr. Ojeda-Burgos has received grants for assistance to medical meetings from Pfizer, Merck Sharp & Dohme, Gilead, Janssen, and Angelini; and has been paid as a speaker in medical meetings from Janssen, Gilead, and Merck Sharp & Dohme. Dr. Miró has received consulting honoraria and/or research grants from Angelini, Bristol-Myers Squibb, Contrafect, Genentech, Gilead Sciences, Merck Sharp and Dohme, Medtronic, Novartis, Pfizer, and ViiV. All authors (including Dr. Ojeda-Burgos and Dr. Miró) have reported that they have no interest conflicts to disclose related to the contents of this paper."

3. One of the noted authors is a group or consortium [GAMES investigators]. In addition to naming the author group, please list the individual authors and affiliations within this group in the acknowledgments section of your manuscript. Please also indicate clearly a lead author for this group along with a contact email address.

Reviewers' comments:

Reviewer's Responses to Questions

**Comments to the Author**

1. Is the manuscript technically sound, and do the data support the conclusions?

Reviewer #1: Yes

Reviewer #2: Yes

2. Has the statistical analysis been performed appropriately and rigorously? 

Reviewer #1: Yes

Reviewer #2: Yes

3. Have the authors made all data underlying the findings in their manuscript fully available?

Reviewer #1: Yes

Reviewer #2: Yes

4. Is the manuscript presented in an intelligible fashion and written in standard English?

Reviewer #1: No

Reviewer #2: Yes

5. Review Comments to the Author

**Reviewer #1: **Dear the authors. Of the manuscript entitled. Clinical presentation, microbiology, and prognostic factors for prosthetic valve endocarditis. Lessons learned from a large perspective registry. Thank you for writing this manuscript which elaborated on different aspects of Prosthetic Valve Endocarditis Presentation, Microbiology and Prognostic factors. Through a study that included two periods of time.

My points to consider. Are the following.

1. This manuscript needs intense English language spelling check. And this language issue is apparently present in almost all of the manuscript sections.

2. There are certain terms that need to be really refashioned such as: Injecting drug user In Table 1., Cardiac insufficiency In Table 2. These terms probably should be rewritten in a more professional way.

3. There was a conflicting sentence to mein the result section in Page. 14. Which says that: Comorbidities, Intracardiac complications and septic shock were more frequent in patients treated during the second period of the study While mortality was improved in this period. This result need to be further explained why mortality was less, despite higher complications and comorbidities were there.

4. The authors mentioned that almost 1/3 of the surgically indicated cases who are having a clear indication for surgery were not operated upon. And these cases that were not operated upon had more risk for mortality. Can you elaborate more? Why there was an obvious reluctance to operate on these patients? And was this a trend because of the surgical risk or the surgeons are hesitant to perform these procedures?

5. Can the authors elaborate more about the methods of echocardiographic assessment used in PVE and whether there was any change between the 2 periods of time

6. The conclusion statement need to reflect the authors opinion about the results they get in this study and should be easy to be understood by the reader.

Thank you

**Reviewer #2: **This is a large prospective cohort study in which the authors evaluated the clinical presentation, the microbiology and the prognostic factors of 1354 patients affected of prosthetic valve endocarditis. The authors analysed the data of all patients affected of prosthetic valve endocarditis undergoing and not undergoing a surgical treatment. They excluded correctly the endocarditis on native valve, on pacemaker or ICD and on TAVR. A lot of factors were evaluated and analysed to better understand the characteristics of this population: the survival, the surgical treatment, the time of onset (early or late), the site of prosthetic infection (aortic or mitral) and the study period (2008-2013 or 2014-2020). Moreover, a multivariate analysis of clinical factors associated with in-hospital mortality (with considering and without considering septic shock) were performed. The authors paid particular attention, in the discussion and in the conclusion, to the higher mortality of patients with prosthetic valve endocarditis that presented clear surgical indication and that didn’t undergo cardiac surgery for the higher surgical risk.

The topic of this study is very interesting and has been extensively analysed.

However, there are some points of discussion:

1. The manuscript needs an English revision.

2. I suggest to add in the Table 1 a column with the characteristics of the “Overall population” of patients affected of endocarditis. This could be interesting considering the high number of patients analysed.

3. In the Figure 1, I suggest to remake the graphic designer in order to better understand the patients included and excluded in the analysis.

4. The lines 205-206, namely the description and the final number of patients analysed in the study, should be moved to the “Patients” section.

6. PLOS authors have the option to publish the peer review history of their article (what does this mean?). If published, this will include your full peer review and any attached files.

Reviewer #1: **Yes: **Salah Eldien Altarabsheh

Reviewer #2: No

---

## [Author Response · Author response to Decision Letter 0]

3 Aug 2023

Dr. Antonio Ramos

Infectious Diseases Unit

HU Puerta de Hierro

Majadahonda. Madrid. Spain. 28222 

 29th July 2023

Phone: +34-689 999 333

E-mail: aramos220@gmail.com

Dear Editor 

We would first thank the editor and reviewers for their valuable comments, which have made a substantial contribution to improving the quality of the manuscript: "Clinical presentation, microbiology, and prognostic factors of prosthetic valve endocarditis. Lessons learned from a large prospective registry" [PONE-D-23-16421. In the following lines we will try to respond, to the best of our ability, to each of the referees' recommendations. We hope that the answers provided and the changes made to the manuscript will meet with the approval of the editor and reviewers. 

Comments to the Author

Reviewer #1: 

1. This manuscript needs intense English language spelling check. And this language issue is apparently present in almost all of the manuscript sections. A revision of the English language has been carried out in order to improve grammatical correctness and text comprehension.

2. There are certain terms that need to be really refashioned such as: Injecting drug user In Table 1., Cardiac insufficiency In Table 2. These terms probably should be rewritten in a more professional way. We appreciate the reviewer's suggestion. The tables have been revised so that the above and other variables are more appropriately expressed. 

Reviewing the number of articles that included the different ways of expressing this harmful habit: People Who Inject Drugs, Injecting Drug Users, intravenous Drug Users, we have observed that the last one is the most preferred by the authors. Consequently, we have changed it. Due to the same reason acute renal injury has been replaced by acute renal failure. We have also changed cardiac insufficiency referring to comorbidity for chronic heart failure and heart failure (during hospital admission) for acute heart failure. 

3. There was a conflicting sentence to main the result section in Page. 14. Which says that: Comorbidities, Intracardiac complications and septic shock were more frequent in patients treated during the second period of the study While mortality was improved in this period. This result needs to be further explained why mortality was less, despite higher complications and comorbidities were there. In “Discussion” it is mentioned that the better prognosis of the patients in the second period (in spite of presenting a poor clinical situation in some aspects) could be related to a progressive increase in the quality of the medical care provided to the patients over time. The speed of diagnosis, the better visualization of each patient's pathology and the progressive learning of surgical skills in each participating hospital over time are variables that are not easy to ascertain and are probably some of the causes of this finding. To a great degree, we do not know if there is any underlying cause responsible for this prognostic improvement. The discussion has been modified and a bibliographic citation has been added describing a lower mortality over time in patients with PVD. [Perrotta S, Jeppsson A, Fröjd V, Svensson G. Surgical Treatment of Aortic Prosthetic Valve Endocarditis: A 20-Year Single-Center Experience. Ann Thorac Surg. 2016 Apr;101(4):1426-32] .

4. The authors mentioned that almost 1/3 of the surgically indicated cases who are having a clear indication for surgery were not operated upon. And these cases that were not operated upon had more risk for mortality. 

Can you elaborate more? Table 5 shows a comparison of several variables that allow visualizing the differences between these two groups of patients.

Why there was an obvious reluctance to operate on these patients?And was this a trend because of the surgical risk or the surgeons are hesitant to perform these procedures? Failure to perform surgery in patients with an indication is a very relevant problem, but unfortunately it is quite frequent one. We think that there is not a single reason for this fact, but several of them. Advanced age, important comorbidity, the poor prognosis some patients (despite surgery) due to a bad clinical condition of and the degree of experience and professional competence of the different surgical teams are factors related to this problem. We think that in most cases it was assumed that when surgery is rejected, it would not be of any benefit to the patients because of, for example, septic shock or neurological complications.

In "Results" the reasons given for not having performed surgery in patients with a surgical indication are shown. In the "Discussion" section we have tried to go deeper into this problem suggesting possible strategies to reduce the number of patients denied surgery. Several comments are presented in relation to the comparison of the clinical characteristics of the patients accepted and rejected for surgery.

5. Can the authors elaborate more about the methods of echocardiographic assessment used in PVE and whether there was any change between the 2 periods of time We are grateful for the reviewer's suggestion. We have added to Table 3S the number of patients in each period in whom transthoracic echocardiography (TTE) and transesophageal echocardiography (TEE) were performed and the number of patients in whom each of these two imaging tests was the only echocardiography performed. There were more cases of patients with TEE performed in the second period. The model of echograph used was not specifically recorded in the database, but the resolution of these technologies, logically, may have progressively improved during the study, although it is not possible to pinpoint the specific changes given the large number of participating hospitals.

6. The conclusion statement need to reflect the authors opinion about the results they get in this study and should be easy to be understood by the reader. We have tried to reflect the authors' opinion on the results in the conclusions section by modifying its structure. We have also tried to improve the reader's understanding.

Reviewer #2: This is a large prospective cohort study in which the authors evaluated the clinical presentation, the microbiology and the prognostic factors of 1354 patients affected of prosthetic valve endocarditis. The authors analysed the data of all patients affected of prosthetic valve endocarditis undergoing and not undergoing a surgical treatment. They excluded correctly the endocarditis on native valve, on pacemaker or ICD and on TAVR. A lot of factors were evaluated and analysed to better understand the characteristics of this population: the survival, the surgical treatment, the time of onset (early or late), the site of prosthetic infection (aortic or mitral) and the study period (2008-2013 or 2014-2020). Moreover, a multivariate analysis of clinical factors associated with in-hospital mortality (with considering and without considering septic shock) were performed. The authors paid particular attention, in the discussion and in the conclusion, to the higher mortality of patients with prosthetic valve endocarditis that presented clear surgical indication and that didn’t undergo cardiac surgery for the higher surgical risk.

The topic of this study is very interesting and has been extensively analysed.

However, there are some points of discussion:

1. The manuscript needs an English revision. A revision of the English language has been carried out to try to improve its grammatical correction.

2. I suggest to add in the Table 1 a column with the characteristics of the “Overall population” of patients affected of endocarditis. This could be interesting considering the high number of patients analysed. A column has been added with the total number of patients as suggested by the reviewer.

3. In the Figure 1, I suggest to remake the graphic designer in order to better understand the patients included and excluded in the analysis. Figure 1 has been modified for a better understanding of the patients included and excluded in the analysis.

4. The lines 205-206, namely the description and the final number of patients analysed in the study, should be moved to the “Patients” section. 

We agree with the reviewer that this information could be added to "Methods". However, and given that the results include a comparison of patients with IE on native valve with patients with IE on prosthetic valve, we consider that it might be more appropriate to keep under "Results" the total number of patients with IE and the number of patients with each type of infection (native or prosthetic).

We would like to thank you once again. 

Sincerely yours

Antonio Ramos

---

## [Decision Letter · Decision Letter 1]

21 Aug 2023

Clinical presentation, microbiology, and prognostic factors of prosthetic valve endocarditis. Lessons learned from a large prospective registry.

PONE-D-23-16421R1

Dear Dr. Ramos-Martínez,

We’re pleased to inform you that your manuscript has been judged scientifically suitable for publication and will be formally accepted for publication once it meets all outstanding technical requirements.

Kind regards,

Redoy Ranjan, MBBS, MRCSEd, Ch.M., MS (CV&TS), FACS

Academic Editor

PLOS ONE

**Additional Editor Comments (optional): **The authors are thanked for this submission to PLOS ONE. After a critical external peer review by two experts and considering the overall reviewers' comments and authors' responses, your manuscript meets PLOS ONE's publication criteria; fulfils the methodological rigour and ethical standards.

**Review Comments to the Author**

Reviewer #1: Dear the authors

Thank you for taking in consideration all the reviewers comments

I am satisfied with the current version of the manuscript

Reviewer #2: Thank you for addressing my comments and for the opportunity to review this paper. There are no further comments from my side.

---

## [Editor Report · Acceptance letter]

30 Aug 2023

PONE-D-23-16421R1 

Clinical presentation, microbiology, and prognostic factors of prosthetic valve endocarditis. Lessons learned from a large prospective registry. 

Dear Dr. Ramos-Martínez:

I'm pleased to inform you that your manuscript has been deemed suitable for publication in PLOS ONE. Congratulations! Your manuscript is now with our production department. 

Kind regards, 

on behalf of

Dr. Redoy Ranjan 

Academic Editor

PLOS ONE